# Targeted Therapies for the Neurofibromatoses

**DOI:** 10.3390/cancers13236032

**Published:** 2021-11-30

**Authors:** Lauren D. Sanchez, Ashley Bui, Laura J. Klesse

**Affiliations:** 1Department of Pediatrics, Division of Neurology, UT Southwestern Medical Center, Dallas, TX 75235, USA; Lauren.Dengle@UTSouthwestern.edu; 2Department of Pediatrics, Division of Hematology/Oncology, UT Southwestern Medical Center, Dallas, TX 75235, USA; Ashley.Bui@UTSouthwestern.edu

**Keywords:** neurofibromatosis, low grade glioma, plexiform neurofibroma, vestibular schwannoma

## Abstract

**Simple Summary:**

The neurofibromatoses—neurofibromatosis type 1, neurofibromatosis type 2, and schwannomatosis—are genetic tumor predisposition syndromes in which affected patients are at risk for the development of nerve-associated central and peripheral tumors. Patients often develop multiple tumors which can result in significant symptoms and morbidity. Treatment of the tumors associated with these disorders has evolved over the past decade, including significant work focused on inhibition of the signaling dysregulation and symptom minimization. This review outlines the most common tumor types associated with each of these syndromes and the current progress in therapeutic options.

**Abstract:**

Over the past several years, management of the tumors associated with the neurofibromatoses has been recognized to often require approaches that are distinct from their spontaneous counterparts. Focus has shifted to therapy aimed at minimizing symptoms given the risks of persistent, multiple tumors and new tumor growth. In this review, we will highlight the translation of preclinical data to therapeutic trials for patients with neurofibromatosis, particularly neurofibromatosis type 1 and neurofibromatosis type 2. Successful inhibition of MEK for patients with neurofibromatosis type 1 and progressive optic pathway gliomas or plexiform neurofibromas has been a significant advancement in patient care. Similar success for the malignant NF1 tumors, such as high-grade gliomas and malignant peripheral nerve sheath tumors, has not yet been achieved; nor has significant progress been made for patients with either neurofibromatosis type 2 or schwannomatosis, although efforts are ongoing.

## 1. Introduction

Recent neurofibromatosis (NF)-focused clinical trials and consensus guidelines have highlighted the unique behavior and management needed for tumors which arise in this group of rare, genetic tumor predisposition syndromes. The neurofibromatoses are a group of three distinct genetic disorders which predispose one to the development of peripheral and central nervous system tumors and include neurofibromatosis type 1, neurofibromatosis type 2, and schwannomatosis. Although most of the tumors which arise are benign in nature, they are associated with significant clinical morbidity and the risks of malignant progression and mortality. No cure for any of the three NF disorders has been identified, and until recently, surgical resection of symptomatic tumors remained the only standard therapy. Treatment of the tumors associated with these disorders has evolved over the past decade, including significant work focused on inhibition of the signaling dysregulation and symptom minimization. Coordinated, multicenter trials have advanced the current understanding of the manifestations of NF and moved potential medical therapies more quickly through testing. In this review, we identify the most common tumors which arise in these patient populations and highlight recent therapeutic trials which leverage what is known about the aberrant cell signaling, particularly for neurofibromatosis type 1 and neurofibromatosis type 2.

## 2. Neurofibromatosis Type 1

Neurofibromatosis type 1 (NF1) is a common tumor predisposition syndrome, with an incidence estimated to be about 1 per 3000 individuals [1]. The most notable effects of NF1 involve the nervous system, but it is a multisystem disorder with a wide range of clinical manifestations. Individuals with NF1 typically manifest characteristic features, including café-au-lait macules, intertriginous freckling, and Lisch nodules (hamartomas of the iris) in childhood, but the disorder may not be recognized until later in many patients. Other common clinical manifestations include learning disabilities, characteristic osseous lesions, and a combination of benign and malignant central and peripheral nervous system tumors. It is from the most common tumor type in NF1, neurofibromas, that the syndrome derived its name. Neurofibromas are often grouped into two distinct entities, cutaneous neurofibromas and plexiform neurofibromas (PNs). Cutaneous neurofibromas are discrete dermal lesions associated with a single peripheral nerve, affecting almost all patients with NF1 by adulthood. PNs, however, are lesions involving multiple nerve fascicles or branches associated with significant morbidity and a risk of transforming into highly aggressive sarcomas known as malignant peripheral nerve sheath tumors (MPNSTs). Following neurofibromas, gliomas are the second most common tumor type in NF1. Optic pathway gliomas are the most common of these, but other gliomas, including high-grade gliomas, are also encountered. Individuals with NF1 are also at an increased risk for non-nervous system tumors, including but not limited to leukemia, pheochromocytoma, breast carcinoma, and gastrointestinal stromal tumors [2,3,4,5,6,7,8,9].

NF1 is an autosomal dominant disorder caused by inherited or de novo germline mutations of the *NF1* tumor suppressor gene located at chromosome 17q11.2, in most cases leading to a loss of function of the *NF1* gene product neurofibromin [1]. Neurofibromin, a large multifunctional protein, is expressed ubiquitously, especially during development, but it is found at highest levels in neurons, Schwann cells, and oligodendrocytes [10,11]. It is a GTPase-activating protein which acts as the common upstream molecule of several pathways, including the Ras/Raf/MEK/ERK signaling pathway and the interrelated mammalian target of rapamycin (mTOR) signaling pathway [12]. Through these pathways, among others, neurofibromin is involved in the regulation of fundamental cellular processes such as cell proliferation and growth. The loss of neurofibromin results in loss of regulation of these functions and allows increased cell growth and tumorigenesis [13]. Figure 1 summarizes the signaling cascades implicated in NF1 tumorgenicity. It is important to note that the specific pathway or mechanism through which this occurs varies depending on the tissue involved (i.e., in leukemic cells versus optic pathway gliomas) [14]. Furthermore, depending on the specific manifestation or tumor type, there may be additional requirements of the progenitor cells and microenvironment, including the need for a second hit and/or haploinsufficiency. For example, from genetically engineered mice, we have learned that haploinsufficiency of Schwann cells is not sufficient for the formation of a PN. A second hit causing inactivation of the *NF1* gene in Schwann cells is required, and haploinsufficiency of other cells in the surrounding microenvironment is additionally required for tumor formation [15].

These nuances have important implications for the treatment of the various manifestations of NF1. With a wide array of disease manifestations and the constellation of tumor types with varying pathogenesis, there is no singular treatment available. Management of the disorder has been mainly focused on routine screening and surveillance, with focused treatment of individual complications as they arise. Specifically, treatment of NF1-associated tumors with more conventional chemotherapy agents has been met with varying degrees of success. With improved understanding of the molecular pathogenesis of NF1 gained over the past decade, targeted treatments of tumors in NF1 have shown encouraging success. Here we will discuss the molecularly targeted approaches to some of the most common or clinically important tumors of NF1.

### 2.1. Plexiform Neurofibromas

PNs occur in up to 50% of people with NF1 [16,17] and occur very rarely in absence of the disorder [18]. They appear to be congenital and tend to present and exhibit their most rapid growth during childhood [19]. PNs are benign peripheral nerve sheath tumors that grow along the length of nerves, often as distinct tumor masses, and can involve multiple fascicles and branches of a nerve [17]. Though histologically benign, PNs tend to be diffuse and infiltrative and can result in significant complications and morbidities, such as pain, impaired motor function, and disfigurement [20,21]. PNs also carry a risk of malignant transformation to the highly aggressive sarcoma MPNST, the leading cause of mortality and reduced life expectancy of patients with NF1 [22].

Histologically, PNs are composed of neoplastic Schwann cells lacking *NF1* gene expression and other cellular and noncellular components, such as *NF1*+/− fibroblasts, perineural cells, and mast cells embedded in a rich mucosubstance collagen matrix [23]. Loss of *NF1* gene expression in the neoplastic Schwann cells, subsequent impairment of neurofibromin-dependent Ras inactivation, and resultant Ras pathway dysregulation are the main cause of tumorigenesis in PNs [24]. However, this is not the complete story. Haploinsufficiency of other cells in the microenvironment, such as fibroblasts and mast cells, also contributes to the pathogenesis of these tumors [25,26]. Additional factors such as increased expression of growth factors and growth factor receptors may also impact tumorigenesis and growth [27,28].

Treatment options for PNs have historically been limited. Conventional cytotoxic chemotherapeutic agents have little effect on these slow-growing tumors. Given the risk of malignant transformation of the PN and the underlying predisposition to cancer in patients with NF1, radiation therapy is generally avoided [29,30]. Treatment of symptomatic lesions therefore has predominantly relied upon surgical debulking. Complete excision of a PN can be curative and this may be rendered increasingly possible via novel techniques such as surgery performed under fluorescein guidance as reported by Vetrano and colleagues [31]. Complete excision, however, is often not feasible due to extensive growth and invasion into surrounding tissues and the nerve itself, resulting in significant post-surgical complications. Tumor recurrence after surgery is also common [32]. The need for additional treatment options for these lesions has been long recognized. Early medication trials using nonspecific agents such as antihistamine, anti-inflammatory, antifibrotic, or antiangiogenic agents to impair the biological processes integral to PN progression had limited success [33,34,35,36,37,38]. More recently, pegylated interferon has demonstrated better results, with one study demonstrating doubling of the time to progression (TTP) in patients with active PNs compared to the placebo group [39]. However, the utility of interferon is limited by often intolerable side effects, and ideally, agents inducing a tumor response rather than just extending the time to progression are sought.

Continued advances in the understanding of the biology and molecular pathogenesis of PNs has led to molecularly targeted treatment approaches. While a marginal response or no response was documented in earlier trials, more recent interventions have been more successful. A first approach was targeted specifically at inhibition of Ras signaling. Ras is a GTPase which serves an integral role in normal cell survival, proliferation, and differentiation by transducing cell surface receptor responses to intracellular signaling molecules [40]. Dysregulation of the Ras signaling pathway is implicated in many tumor types. In a properly functioning cell expressing *NF1*, the *NF1* gene product neurofibromin functions as a negative regulator of Ras by accelerating hydrolysis from the active form Ras-GTP to the inactive form Ras-GDP. In NF1, loss of *NF1* expression leads to constitutive activation of the Ras pathway, which leads to tumorigenesis [41]. Blockade of this aberrant Ras signaling, therefore, is a rational target in the treatment of PNs [42]. Ras proteins in the Ras signaling pathway downstream of the defunct neurofibromin protein in NF1 require post-translational modification by farnesylation to be biologically active. The use of tipifarnib, a farnesyl transferase inhibitor, [43] in a randomized, double-blinded study treating children and young adults with NF1-related progressive PNs demonstrated tolerability of the drug but did not significantly increase TTP compared to placebo [44]. Notably, tipifarnib has been shown to effectively inhibit H-Ras farnesylation, but not other Ras proteins, N-Ras and K-Ras, which can undergo an alternate lipid modification [45]. In NF1-associated tumors, K-Ras is thought to be the predominant isoform involved, potentially underlying the poor clinical result in this trial.

An alternative signaling pathway that has been targeted is the mammalian target of rapamycin (mTOR) pathway, also demonstrated to be integral to cell survival, proliferation, and differentiation [46]. Akin to its role in the Ras pathway, neurofibromin plays a role in regulation of the mTOR pathway, and the mTOR pathway is constitutively activated in *NF1*-deficient cells and tumors [47]. mTOR signaling is upregulated in many cancers, and the mTOR kinase central to the pathway has been a successful target of other cancer treatments, including in the treatment of subependymal giant cell astrocytomas (SEGAs) in another Ras pathway disorder, tuberous sclerosis complex (TSC) [48]. mTOR was named for its susceptibility to the macrolide compound rapamycin, as rapamycin, in complex with another protein, inhibits the function of mTOR [49]. Rapamycin (also known as sirolimus) was trialed in NF1 for both non-progressive and progressive inoperable NF1-associated PNs but had limited success (NCT00634270). In patients with non-progressive PNs, no objective radiographic responses were demonstrated [50]. In patients with progressive PNs, a modest increase of 4 months in TTP was noted with without significant or frequent toxicity in a subset of patients [50]. Everolimus, another mTOR inhibitor, and pexidartinib, an inhibitor of a tyrosine kinase in the mTOR signaling pathway, have also been studied without much success [51,52]. Everolimus, in particular, was associated with significant adverse effects [53].

Conversely, the use of mitogen-activated protein kinase (MAPK) inhibitors has demonstrated significant responses and activity—a clear major advancement in clinical treatment for these patients. In the Ras/MAPK pathway, activation of Ras results in sequential activation of RAF kinase, MEK, and MAPK (ERK), respectively. MAPK acts as a regulator for several transcription factors, thereby acting as a regulator of the transcription of genes important in the cell cycle [42]. MEK inhibition has excitingly proven to be beneficial in the treatment of NF1-associated PNs. In 2016, a phase 1 study reported initial evidence of volumetric shrinkage of PNs in children who received the selective MEK inhibitor, selumetinib [54]. This was followed with a phase 2 study, published in 2020, which confirmed the previously reported radiographic response of >20% volumetric shrinkage. In addition to a sustained tumor response for 1 year or longer, 68% of the patients had some degree of clinical improvement in at least one PN-related complication, such as pain or limitation in physical functioning [55]. The therapy had an acceptable safety profile, with the most common adverse events including gastrointestinal symptoms (nausea, vomiting, and diarrhea), an asymptomatic increase in the creatine phosphokinase level, acneiform rash, and paronychia [55,56]. In response to these studies, selumetinib was granted FDA approval in 2020 for patients ≥2–18 years old with NF1 and symptomatic, inoperable PNs. The success of this therapy has spurred additional trials using selumetinib in adult patients with NF1-associated PNs, and trials utilizing alternative MEK inhibitiors such as trametinib, binimetinib, and mirdimetinib. These trials, some completed and some ongoing, have had subtle differences and varying degrees of efficacy, but as a group have shown remarkable success [57]. Limitations to the usage of MEK inhibitors in the treatment of PNs are not inconsequential and include a minimal tumor response often appreciated by conventional imaging, no complete responses obtained, a third of patients do not respond, and administration of the drug appears to be required over extended time periods, as PNs often become progressive after cessation. Therefore, although very exciting as a new medical therapy for NF1-associated PNs, ongoing work is clearly needed.

The use of tyrosine kinase inhibitors, specifically cabozantinib, has demonstrated activity as well and warrants ongoing investigation. This approach targets the critical role of the tumor microenvironment (TME) in PN formation. A previous study using imatinib mesylate, another tyrosine kinase inhibitor, to treat PNs showed modest success, but primarily in small tumors [58]. Cabozantinib, on the other hand, has demonstrated more success. Cabozantinib is a multiple tyrosine kinase inhibitor which, in preclinical and translational studies, modulated key kinases in the TME in *Nf1*-mutant mice and reduced the PN tumor burden in these mice [59]. Based on this rationale, adolescents and adults with NF1 and progressive or symptomatic unresectable PNs were enrolled in a phase 2 clinical trial for treatment with cabozantinib. In this study, 42% of participants achieved a partial response (PR, defined as ≥20% reduction in tumor lesion volume as assessed by MRI), and patients with PR had significant reductions in tumor pain intensity and pain interference in daily life. The medication was reasonably well tolerated, though a significant portion of patients did discontinue it due to low-grade adverse effects [59]. The response rate in this trial was the best rate seen thus far in adults, and future data from treatment of pediatric patients with cabozantinib and adult patients with selumetinib may better allow for comparisons of response to the two agents. Overall, these trials demonstrate monumental potential in another class of agents that are efficacious at increasing TTP, decreasing tumor size, and inducing a clinical benefit in patients with previously poorly remediable tumors. Combination strategies could potentially have even further benefits; investigation into the use of cabozantinib in combination with an MEK inhibitor is planned. With the knowledge gained through these endeavors over the past decade and the success of these two molecularly targeted approaches, a promising era in the treatment of NF1-associated PN has been entered.

### 2.2. Malignant Peripheral Nerve Sheath Tumors

PNs develop from any peripheral nerve branch or bundle and carry a risk of malignant transformation into an aggressive soft tissue sarcoma, or malignant peripheral nerve sheath tumor (MPNST) [60]. MPNST occurs in patients with and without NF1 but may be associated with worse outcomes in the former group [61]. In NF1, these high-grade tumors can occur sporadically but more often arise from a pre-existing PN. Patients with NF1 have an approximately 10–15% lifetime risk of MPNSTs [61,62,63]. MPNST is frequently associated with distant metastatic disease and local recurrence [64]. Risk factors for developing MPNST include whole body PN burden, presence of nodular or atypical lesions, prior radiation therapy, and *NF1* microdeletions [61].

The clinical presentation of MPNST is heterogeneous depending on tumor size and location, but patients with severe or refractory pain, new neurologic deficits, rapid growth, or hardening consistency of an existing PN warrant radiologic evaluation [62]. While MRI can be used to determine the location and extent of the tumor and may indicate changes, ^18^F-fluorodeoxyglucose (FDG)-PET scan has been demonstrated to better distinguish between benign versus malignant tumor, as MPNST will demonstrate increased FDG uptake [62,63]. This can be particularly useful in the setting of a smaller malignant transformation in a larger PN to help direct surgical intervention.

In addition to the prerequisite biallelic inactivation of *NF1*, other somatic alterations contribute to the cancerous behavior of these tumors. Inactivation of other tumor suppressor genes, such as *TP53*, *CDKN2A/B*, *PTEN*, and the polycomb repressor complex 2 (PRC2, containing *EED* and *SUZ12*), and the amplification of growth-promoting genes, such as *EGFR* and *PDGFR*, have been documented [65,66]. Interestingly, alterations in *NF1*, *CDKN2A/B*, and the PRC2 genes are frequently found concurrently in non-NF1-associated MPNST as well, supporting the notion that sequential inactivation of these genes drives the malignant evolution of tumors [66].

Ultimately, disease progression from PN to MPNST involves the stepwise acquisition of additional genetic mutations and chromosomal rearrangements. Atypical neurofibromas (ANs) are indolent, premalignant, nodular lesions that develop from and/or within PNs. A subset of these tumors can be further classified as atypical neurofibromatous neoplasms of uncertain biologic potential (ANNUBP), which share more histologic features with MPNST [67,68]. Deletion in the *CDKN2A/B* gene appears to be the first step in disease progression [69]. Mutations in *SMARCA2* have also been identified, though their role in malignant transformation is not yet known. While not all ANs will fulfill their malignant potential, the approach to treatment involves surgical resection followed by clinical surveillance given the significant risks associated with MPNST [60,70].

Therapeutic options for MPNST remain limited. To date, surgery is the only curative option, though similarly to PNs, this is often limited by the invasive nature of the tumor and frequent involvement of vital surrounding structures [61,66,71]. The goal of surgical resection is complete excision with negative margins if feasible. Significant postoperative morbidity, however, with loss of sensory and/or motor function, particularly when major nerves are involved, is often seen and therefore supports a more cautious surgical approach that removes as much tumor as safely possible while attempting to preserve neurological functions and thus, quality of life [71,72]. Radiation, which again is typically avoided in patients with cancer predisposition syndromes, can be used for local control though it has been observed to delay the time to disease recurrence but not death [62]. Historically, MPNST has not demonstrated a clear response to systemic chemotherapy, but for certain individuals, neoadjuvant agents such as doxorubicin and ifosfamide, which are often used in the treatment of other sarcomas, may be used to shrink the tumor and optimize the feasibility of a gross total surgical resection [64].

Given its aggressive nature, MPNST is frequently associated with difficulty in achieving local control, and thus a poor prognosis. It represents a major cause of morbidity and mortality in patients with NF1 [62,64]. As with PNs and other clinical manifestations of NF1, increased understanding of molecular drivers in MPNST and expanded use of targeted therapies such as MEK inhibitors have paved the way for new therapeutic approaches. Current clinical trials for MPNST include a phase 2 study combining selumetinib and sirolimus for MEK and mTOR inhibition, respectively (NCT03433183). There are several studies combining mTOR inhibitors with other forms of targeted therapy (NCT01661283, NCT02008877, NCT02584647, NCT02601209). The emerging use of immunotherapy has also been integrated into early phase studies, including immune checkpoint inhibitor therapy (nivolumab, a PD-1 inhibitor, and ipilimumab, a CTLA-4 inhibitor [NCT02834013]; pembrolizumab with APG-115 [NCT03611868]); chimeric antigen receptor T-cells targeting EGFR (NCT03618381); and vaccine therapy (NCT02700230). Continued advances in genetic and molecular profiling are expected to provide new insights into tumor development and hopefully, translate to improved methods of detection, diagnosis, and treatment.

### 2.3. Low-Grade Gliomas

Low-grade gliomas (LGGs) are the most common intracranial tumors in NF1 [73]. The vast majority of these are pilocytic astrocytomas, World Health Organization (WHO) grade I tumors. They can occur almost anywhere in the brain but are most commonly found in the optic pathways or the brainstem [73]. Optic pathway gliomas (OPGs) can affect the optic nerves, chiasm, post-chiasmatic tracts, or radiations, and they can extend to the hypothalamus. They occur in 15–20% of patients with NF1 and typically present prior to 7 years of age [74]. Compared to the general population, NF1-associated OPGs tend to have a more benign course [75]. At least half of OPGs remain asymptomatic [76]. When symptoms do occur, patients can present with strabismus, ptosis, proptosis, pain, pupillary changes, vision changes, hypothalamic disturbances, and rarely, hydrocephalus. In the absence of symptoms, or even in mildly symptomatic cases, intervention for these tumors is most often unnecessary. There are no clear prognostic features to guide when intervention is needed, but suggested risk factors for progression include female sex, age of presentation less than 2 years or greater than 8–10 years, or tumor location in the post-chiasmatic optic pathway [77,78,79].

Like PNs and other solid tumors, OPGs are products of neoplastic cells with contributions from the TME. NF1-associated OPGs arise from glioma stem cells and astrocytes with bi-allelic inactivation of the *NF1* gene in a microenvironment of stromal cells including microglia, neurons, and endothelial cells haploinsufficient for *NF1*. As with PNs, these haploinsufficient cells are required for tumorigenesis [80]. It is the lack of negative regulation of the Ras signaling pathway by neurofibromin and aberrant signal transduction through the Raf/MEK/MAPK and mTOR signaling pathways, which underlies the development of OPGs [47,81].

Given their indolent course and low risk of progression, observation of OPGs in asymptomatic patients is often the preferred approach to management. Screening MRIs are not recommended, as, in the absence of clinical findings, detection of OPGs on MRI is unlikely to change management [82]. Standard screening in patients with NF1 for OPGs includes annual eye exams by an experienced pediatric ophthalmologist for all patients less than 10 years of age, then at least every two years until 18 years of age [83]. In addition, children with NF1 should undergo yearly height and weight measurements screening for precocious puberty or other evidence of hypothalamic dysfunction [83]. MRI is indicated in children with screening findings suggestive of OPG or potentially in children in whom reliable screening cannot be performed. Once an OPG is identified, increased ophthalmologic evaluation and MRI evaluations are indicated, though there is no consensus on timing interval [83]. MRI progression and visual outcomes do not clearly correlate; therefore, radiographic progression alone is often not an indication for therapy in NF1 patients [84]. When intervention is required, options remain focused on chemotherapy. Surgery may have a role when vision has already been lost in the affected eye or to treat specific ophthalmologic issues such as corneal exposure, but otherwise surgery is avoided as the goal of therapy is to maintain vision [83]. Radiation therapy to the tumor in this population is generally avoided due to the risk of secondary malignancy and the risk of moyamoya syndrome in patients with NF1 [85]. When indicated, first-line therapy for NF1-associated OPG is typically conventional cytotoxic chemotherapy, most often with carboplatin and vincristine [86]. Chemotherapy is effective at halting tumor progression in the majority of cases, but this may come with substantial systemic side effects [87]. Excitingly, OPGs have also shown encouraging response to MEK inhibition. In a phase 2 study treating patients with NF1-associated pediatric LGGs with selumetinib, 40% of patients achieved a sustained partial response, 96% of patients had 2 years of progression free survival, and the medication was overall well tolerated during the study [88]. Given this robust result, a phase 3 non-inferiority trial by the Children’s Oncology Group is now underway, comparing carboplatin and vincristine to selumetinib monotherapy (NCAT03871257).

### 2.4. High-Grade Gliomas

While LGGs affecting the optic pathway and other intracranial structures predominate, patients with NF1 also have an estimated fifty-fold increased risk of developing high-grade gliomas (HGGs) compared to unaffected individuals [66,89,90]. HGGs, which include glioblastoma, anaplastic astrocytoma, and other histologic subtypes, typically present after the age of 10 years with most cases occurring in adulthood [66,89,90]. These tumors tend to affect the cerebral hemispheres and are more likely to be symptomatic than low-grade tumors [60,91].

Advances in molecular profiling and the overall understanding of tumor biology support the morphologic evolution from LGGs to HGGs with accumulation of additional somatic mutations [92]. In addition to the germline *NF1* mutation, a somatic mutation of the second *NF1* allele is common but not required for tumorigenesis. Based on genomic analysis of 59 NF1-associated glioma samples from pediatric and adult patients, HGGs were frequently found to have higher mutational burdens with abnormalities in *TP53* and *CDKN2A*, along with inactivation of *ATRX*, which appears to correlate with more aggressive clinical behavior [89,90]. DNA methylation is an emerging factor in the classification of brain tumors and potentially in NF1-associated gliomas as well [92]. Notably, hallmark mutations in the *IDH* genes or H3.3 histone genes which are found in sporadic pediatric gliomas have not been identified in NF1-associated HGGs [91,93].

The current treatment approach remains similar to that of sporadic cases, centering around surgical resection, adjuvant radiation, and chemotherapy such as temozolomide [62,91]. Though data are limited, largely by the rarity of the diagnosis, HGGs carry a poor prognosis with an overall survival of 50–60 weeks in adults and only slightly better in children [93,94].

## 3. Neurofibromatosis Type 2

Neurofibromatosis type 2 (NF2) is less common than NF1, affecting 1 in 25,000 individuals worldwide [95]. Individuals with NF2 are at risk for a variety of nervous system tumors including peripheral and central schwannomas, particularly vestibular schwannomas (affecting cranial nerve VIII), multiple meningiomas, and ependymomas. Affected individuals are also at risk for ocular manifestations such as juvenile posterior subcapsular cataracts and epiretinal membranes which can affect vision. Most of the tumors associated with NF2 are histologically benign in appearance but are clearly associated with significant patient morbidity given their number and often persistent growth over time. Malignant transformation of tumors is typically not seen, except in instances where patients have undergone prior irradiation [96,97].

NF2 is the result of a pathologic variant in the *NF2* gene, and the pathogenic variants result in loss of function of merlin, the protein product of the *NF2* gene. A member of the ERM (erzin/radixin/moesin) family of scaffolding proteins, merlin has been implicated in several cell signaling cascades, including the Ras/MAPK, FAK/SRC, PI3K/AKT and the HIPPO signaling cascades (Figure 2). The specific pathways involved in the development of the tumors in NF2 have not been clearly delineated. Loss of heterozygosity of *NF2* is required for tumor development, in keeping with its known tumor suppressor function, but other cooperating mutations may be necessary [98]. Patients with truncating mutations in the *NF2* gene often present with more severe disease at younger ages while patients with missense mutations will often have a milder course with tumors which are slower to progress [99]. NF2 is inherited in an autosomal dominant manner, however, 50% of newly diagnosed patients represent de novo mutations. Of these proband patients, a significant number will be mosaic for the *NF2* pathologic variant and may not be able to be identified by blood genetic testing even when they met the diagnostic criteria [100]. The Manchester Criteria for NF2 remains the standard for clinical diagnosis and was most recently updated in 2017 to exclude *LZTR1* pathologic variants [101]. The clinical diagnostic criteria include bilateral vestibular schwannomas (prior to age 70); or a known first-degree family member with NF2 and either a unilateral vestibular schwannoma or two or more meningiomas, cataracts, schwannomas, or cerebral calcifications. Alternatively, the diagnosis can also be made with a documented *NF2* pathologic variant along with either a unilateral vestibular schwannoma or two other distinct tumors. Surgical resection of the tumor, when feasible, remains the mainstay of therapy for the majority of patients with NF2, with the goal of minimizing tumor-associated symptoms. Given the work to identify the signaling cascades involved in the development of NF2-associated tumors, however, a number of clinical trials aimed at identifying a potential medical therapy have been undertaken with varied success.

### 3.1. Vestibular Schwannomas

Vestibular schwannomas are the most common intracranial tumor in patients with NF2, affecting up to 90% of individuals and are a significant cause of morbidity. Bilateral vestibular schwannomas are classically associated with a diagnosis of NF2, although bilateral disease may not be present at diagnosis in all patients [102]. Vestibular schwannomas arise from either branch of cranial nerve VIII and can lead to hearing loss, vestibular dysfunction, facial nerve palsies and ultimately brainstem compression. Patients with NF2 have been demonstrated to have numerous tumor nodules along the nerve, indicating that vestibular schwannomas in this population are likely composed of multiple tumor nodules instead of one discrete tumor, thereby complicating surgical resection. The size of the vestibular schwannoma does not correlate with hearing loss, supporting alternative mechanisms besides tumor compression of the nerve as the etiology of hearing loss [103,104]. Overall, the primary goal of therapy is to attempt to prolong functional hearing for as long as feasible. Therefore, surgical resection of NF2-associated vestibular schwannomas is usually considered in the setting of hearing sparing surgery, for progressive tumors once hearing has been lost or in the setting of any brainstem compression [105]. The use of radiation therapy (radiosurgery), a common therapy for sporadic vestibular schwannomas, has declined in patients with NF2. Although radiation therapy does result in tumor growth control, it has been associated with a small risk of malignant transformation [97] and poor hearing outcomes with less than half of patients demonstrating preserved hearing at 5 years [106,107]. At this time, there are limited data on the use of proton-based radiation therapy for NF2-associated vestibular schwannomas, although it is not likely to be significantly different than conventional or radiosurgery-based approaches in terms of hearing preservation. Given these limited options, the identification of medical therapy which cannot only control tumor growth but preserve hearing function has been a high priority for the NF2 care community. Conventional chemotherapy has not been demonstrated to be effective for NF2-associated vestibular schwannomas, and therefore many trials have focused on inhibition of potential NF2-associated molecular pathways. Table 1 summarizes several recent NF2-associated vestibular schwannoma clinic trials including the agents utilized, primary outcomes, enrollment, and results. Bevacizumab has been overall the most successfully utilized and reported, but several agents have demonstrated some preliminary activity.

One of the earliest signaling cascades to be identified as a potential therapeutic target was the mammalian target of rapamycin complex 1 (mTORC1). Loss of merlin results in activation of mTORC1 signaling and resultant cell growth [108]. Inhibition of mTORC1 by rapamycin resulted in decreased growth of merlin deficient Schwann cells both in vitro and in vivo in preclinical testing [109]. Two phase 2 clinical trials with the rapamycin analog, everolimus, were undertaken. Neither study demonstrated a radiographic response (tumor shrinkage of >20%) or hearing improvement in patients with NF2-associated vestibular schwannomas [110]. Time to radiographic progression of the vestibular schwannomas was improved, however, from a median of 4.2 months to more than 12 months [111]. Subsequent evaluation of tumor tissue from a phase 0 trial with everolimus indicated incomplete inhibition of the target pathway in the tumor even when blood levels were adequate, which may explain the minimal clinical response [112].

Lapatinib, an oral epidermal growth factor receptor (EGFR) and Erb2 inhibitor, has also demonstrated activity in NF2. Lapatinib was identified for potential use for NF2-associated vestibular schwannomas due to increased expression of both EGFR and Erb2 in tumor tissue and inhibited proliferation in schwannoma cell lines [113]. A phase 2 clinical trial of lapatinib in patients with NF2 and progressive vestibular schwannomas was undertaken with a primary tumor response endpoint of >15% volumetric reduction. Twenty-one patients were enrolled with 17 of them evaluable for radiographic response. Four of the 17 (23.5%) had tumor volumetric shrinkage of 15.7–23.9% over the therapy course with one response durable beyond 9 months. Hearing was assessed as a secondary aim and 4 of 13 patients (30%) had improvement in word recognition scores [114]. Overall, lapatinib was well tolerated in this patient population and demonstrated some activity. Although no future trials are currently planned with lapatinib, it remains a potential agent for use in NF2-associated vestibular schwannomas.
cancers-13-06032-t001_Table 1Table 1Recent clinical trials assessing targeted therapy for NF2-associated vestibular schwannomas with intended targets, phases of therapy, endpoints, and references.Drug NameTherapy TargetPhase Trial/Number of Patients EnrolledNotable EndpointsFurther Studies PlannedReferencesErlotinibEpidermal Growth Factor Receptor (EGFR)Phase 2–10 Patients3/10 with minimal radiographic responseNoPlotkin et al., 2010 [115]EverolimusMammalian Target of Rapamycin (mTOR)Phase 2–9 PatientsNo radiographic or hearing responses. Prolonged time to progressionNoKarajannis et al., 2014 [110]; Goutagny et al., 2015 [111]LapatinibEGFR and Erb2Phase 2–21 patients(17 evaluable)4/17 with >15% size reduction4/13 with improved hearingUnclearKarajannis et al., 2012 [114]BevacizumabVascular Endothelial Growth Factor Receptor (VEGFR)Phase 2, multiple studies (>100 patients reported)RR in 41%Hearing improvement in 20%TTP improvementNoLu et al., 2019 [116]; Plotkin et al., 2012 [117]; 2019 [118]; Blakeley et al., 2016 [119]Crizotinib/BrigatinibFocal Adhesion Kinase (FAK1)Phase 2, ongoingVolumetric response as primary aim, hearing secondaryOngoingN/A


Aspirin as a modulator of growth of vestibular schwannomas has conflicting reports of efficacy. Vestibular schwannomas, including NF2-associated vestibular schwannomas, have been noted to express COX-2, and the degree of expression correlates with proliferation, indicating that COX-2 inhibition with aspirin might be a viable therapeutic [120]. Initial retrospective analysis of patients with sporadic vestibular schwannomas on aspirin for other indications indicated slower growth rates for those on aspirin [121]. More recent retrospective studies have not supported this correlation, and currently compelling data to recommend aspirin therapy for patients have not been reported [122,123]. To better define a potential role for aspirin, a phase 2 clinical trial for both NF2 and sporadic tumors is currently ongoing (NCT03079999).

The most successful and well reported medical therapy to date for patients with progressive or symptomatic vestibular schwannomas is bevacizumab. Prior work demonstrated increased vascular endothelial growth factor (VEGF) expression in NF2-associated vestibular schwannomas, and initial studies utilizing bevacizumab demonstrated both decreased tumor volume and hearing improvement in a subset of patients [117,119]. Overall, a recent meta-analysis reviewing eight studies covering 161 patients reported both prolonged time to tumor progression and to hearing loss [116]. Bevacizumab is currently considered the first line medical therapy for NF2-associated vestibular schwannomas in the setting of either hearing decline or tumor progression. Of note, pediatric patients may receive less benefit from bevacizumab than do older patients, and there is no clear benefit to the higher dose of 10 mg/kg than the lower doses of 5–7.5 mg/kg every 3 weeks [118]. As the duration of therapy may need to be prolonged, lower dosing regimens may be preferred as they may have a lower risk of toxicities, particularly renal impairment [124].

Both identification of novel potential therapeutic agents and clinical trials are ongoing for vestibular schwannomas. Two trials are currently ongoing to assess the impact of inhibition of alternative NF2-associated signaling cascade molecules such as focal adhesion kinase 1 (FAK1) and EphA2 by utilizing the FDA-approved ALK inhibitors, crizotinib and brigatinib. Crizotinib was identified via drug screening in a preclinical *NF2*-deficient cell line and xenograph testing [125]. Crizotinib inhibited tumor formation by inhibition of focal adhesion kinase 1 (FAK1) and is current being tested in a phase 2 trial for progressive NF2-associated vestibular schwannomas (NCT04283669). Brigatinib was identified as a potential therapeutic target for both meningiomas and vestibular schwannomas as part of a coordinated high throughput screen and in vivo mouse modeling/xenograph testing [126]. Brigatinib inhibits several tyrosine kinases including EphA2 and FAK1 and is also currently undergoing phase 2 clinical testing (NCT04374305). A second signaling pathway currently undergoing clinical trial evaluation is the Ras/extracellular signal-regulated kinase (ERK) cascade with inhibition of MEK [127]. Loss of merlin has been associated with activation of the Ras/ERK signaling cascade. In a mouse model, inhibition of this activation with MEK inhibitors resulted in decreased growth of schwannoma cells and decreased tumor burden and average tumor size [128]. A phase 2 trial of the MEK inhibitor, selumetinib, for progressive NF2-associated tumors is currently ongoing (NCT03095248).

### 3.2. Meningiomas

Meningiomas are the second most common tumor identified in patients with NF2, found intracranially in approximately 45–80% of patients and in the spinal axis in 20%. The development of multiple meningiomas is a hallmark of the disease [129,130]. Loss of *NF2* or instability of chromosome 22 is a frequent feature of sporadic meningiomas as well [131]. Surgical resection continues to be the mainstay of therapy for progressive or symptomatic tumors. Radiation and radiosurgery have also been utilized, but long-term outcome measures have been limited. As with vestibular schwannomas, therapeutic trials have leveraged known NF2 signaling cascades but with only limited success. Unlike NF2-associated vestibular schwannomas, bevacizumab has been demonstrated to have limited benefit, with only a subset of patients demonstrating radiographic response which appears to be of limited duration [132,133]. Retrospective analysis of 8 patients with meningiomas treated on the lapatinib trial for progressive vestibular schwannomas demonstrated slower volumetric growth rates of the meningioma while on the medication when compared to off therapy [134]. Analysis of 6 meningiomas in patients with NF2 who were treated with everolimus for progressive vestibular schwannomas showed prolonged time to progression of the meningiomas from 5.5 months pretreatment to more than 12 months on therapy. No tumor shrinkage was seen [111]. The combination of bevacizumab and everolimus for recurrent meningiomas, though not specifically in patients with NF2, resulted in stable disease for the majority of patients and performed similarly to bevacizumab alone [135]. The phase 2 CEVOREM trial for recurrent meningiomas utilizing everolimus and octreotide, given the strong expression of the sst2 somatostatin receptor in meningiomas, also included patients with NF2. This study reported a decrease in median growth rates of the meningioma from 16.6% over 3 months prior to study inclusion to 0.02% in the first 3 months and 0.48% in the second [136]. A recent trial utilizing the dual mTORC1/mTORC2 inhibitor AZD2014 for patients with NF2 with progressive or symptomatic meningiomas had a significant number of patients withdraw prior to study completion due to intolerable side effects (NCT02831257). Clearly, further research is necessary for this group of patients.

### 3.3. Ependymomas

Development of ependymomas, particularly in the cervical cord or cervicomedullary junction, is also a common finding in patients with NF2, and multiple ependymomas are found in over 50% of patients [137]. The majority of ependymomas associated with NF2 appear to be asymptomatic and exhibit indolent growth. Therefore, a conservative management approach with observation is usually taken. Symptoms develop in approximately 20% of patients with NF2. In these patients, surgical resection, if feasible without significant morbidity, is the primary therapy approach and may be associated with improved neurologic outcome [138]. For patients with symptomatic tumors and no surgical options, bevacizumab has been reported to improve symptoms and has also been shown to lead to radiographic response in some [139,140]. Given their limited morbidity for the majority of patients, fewer therapeutic clinical trials have focused on NF2-associated ependymomas.

## 4. Schwannomatosis

Schwannomatosis is less common, less well studied, and less well understood compared to NF1 and NF2. The general incidence for schwannomatosis is 1 in 40,000, and most patients are diagnosed in their third and fourth decades of life. Schwannomotosis is characterized by the development of multiple non-intradermal schwannomas, and in 5% of patients, meningiomas. Patients with schwannomatosis often present with neurological symptoms—classically chronic pain, which can be either focal or diffuse. Although there is a phenotypic overlap with NF2, two distinct genes have been identified for schwannomatosis, *SMARCB1* and *LZTR1*. These are inherited in an autosomal dominant fashion but with incomplete penetrance, and less than 20% of patients have a known family history of the disease. At this time, there is no medical therapy for the treatment of the tumors associated with schwannomatosis. Therapy is often focused on treatment of the associated pain, typically with gabapentin and/or tricyclic antidepressants. Surgical resection can also be utilized for uncontrolled pain, with the goal of preserving neurologic function. The size of the schwannoma does not appear to correlate with degree of pain, although patients with *LZTR1* pathologic variants have been reported to have increased pain. Current clinical trials focus on pain management, with one study utilizing tanezumab, a monoclonal antibody to nerve growth factor (NGF), as NGF has been implicated as a mediator of pain (NCT04163419). Hopefully, as research progresses in this poorly understood disease, more therapies will be identified.

## 5. Conclusions

Patients with neurofibromatosis are at clear risk for tumor development and subsequent associated morbidity and mortality. While malignant NF-associated tumors, particularly MPNSTs and HGGs in NF1, exhibit clinical behavior similar to their sporadic counterparts, the benign NF-associated lesions often have a distinct, potentially more indolent natural history. This clinical behavior, coupled with the risk of developing multiple tumors over time, has shifted the focus to symptomatic treatment for the majority of these tumors. When symptomatic, however, therapeutic options remain quite limited. With only one FDA-approved therapy for NF1, no approved therapies for NF2 or schwannomatosis, and significant limitations to conventional therapeutic options in general, novel agents are greatly needed. In this review, we have focused on the identification of molecularly targeted therapies which leverage the known signaling aberrations associated with each of these disorders. Although progress has been made, the need for more tumor-directed therapies with tolerable short and long-term toxicities is clear. Given the complexity of the signaling pathways involved, it is probable that combination therapy will be necessary—although concern arises about therapy-related toxicities given the likely need for prolonged therapy.

Additionally, the NF community has made a more coordinated approach to collaborate and accelerate therapy recommendations and develop more clinical trials. Several consortia have formed to design consensus recommendations and identify targets to streamline NF-focused clinical trials. Since 2006, the Department of Defense, as part of the Congressionally Mandated Research Program, has supported a National Neurofibromatosis Clinical Trials Consortium (NFCTC). The NFCTC is focused on designing and performing clinical trials for NF patients. To date, the NFCTC has undertaken 15 trials: 4 focused on PNs, 3 on LGGs, 4 on MPNSTs, and 2 for NF2-associated vestibular schwannomas. The consortium currently has 15 primary sites and 10 affiliate sites across the United States and one in Australia. The Children’s Tumor Foundation has likewise supported a Neurofibromatosis Preclinical Consortium to consistently evaluate potential therapeutic targets. Utilizing genetically engineered mouse models of tumor specific manifestations of both NF1 and NF2, and appropriate cell lines, the preclinical consortium has focused on screening and testing potential therapeutic options of reach tumor type. To date, mouse (and some zebrafish) models of MPNSTs, PNs, and OPGs have been developed for NF1 while a vestibular schwannoma model has been developed for NF2 [141,142,143]. Although rodent model systems have been extremely beneficial for understanding the mechanisms of tumor formation, they have been less consistent as models for the multiple manifestations of disease or as preclinical tests of potential therapeutic targets. To improve the translatability of findings to clinical use, more recent efforts have focused on development of more relevant animal modeling systems, such as a minipig NF1 model, or on more in-depth analysis of human-derived tumors for target identification [144,145]. Included in these efforts are the Children’s Tumor Foundation developed series of Synodos programs: collaborations between multidisciplinary team members focused on identifying potential therapeutic targets and bringing novel therapeutics forward in an open data sharing approach. Four groups have been formed to date and include (1) a group focused on identification of molecular targets in NF1-associated gliomas by sequencing analysis of patient samples, (2) a group trying to accelerate preclinical testing in NF1 by development of a minipig model of NF1, a closer to human model system, (3) a group trying to identify therapies for NF2-associated complications by utilizing both cell and animal model systems for screening (which identified Brigitinib as a potential therapy), and (4) a group focused on improving therapy for schwannomatosis by undertaking molecular analysis of patient-derived tumors. The overall goal of this program is acceleration of drug discovery and translation to clinical trials.

Given the complexity of NF-associated tumors, including their unique natural histories and indications for therapy, the standard imaging response, i.e., tumor shrinkage, has been recognized as potentially not an ideal representation of agent efficacy or clinical improvement in this patient population. In response, an international group was formed in 2011—the Response Evaluation in Neurofibromatosis and Schwannomatosis (REiNS) International Collaboration, with the goal of identifying improved endpoints, particularly for clinical trial design, which better reflect the clinical need in this patient population. The group currently includes seven working groups, five of which are focused on outcome measures, including tumor imaging, functional, visual, patient reported, and neurocognitive outcomes. The other two groups examine whole body MRI and disease biomarkers. REiNS has published several recommendations regarding NF-associated outcome measures—such as visual acuity for OPGs and volumetric imaging for PNs—for trials to give better consideration to long term clinical benefits [146,147].

Overall, therapeutic approaches to NF-associated tumors cannot simply be modeled after sporadic counterparts, particularly for the benign tumors associated with these disorders. The unique role of the microenvironment in tumor development, the window of risk during development, and the risk for the formation of multiple tumors require a unique clinical response which prioritizes clinical outcomes and symptom management. This need has resulted in numerous clinical trials aimed not only at tumor control but evaluation of clinically relevant outcome measures such as pain, hearing, and functional outcomes. Ongoing research into the aberrant signaling cascades and the interactions of these cascades will continue to identify potential therapeutic targets which may improve patient outcomes. A landmark FDA-approved therapy has been achieved for NF1-associated PNs, but similar success for NF2 and schwannomatosis is still needed.

## Figures and Tables

**Figure 1 cancers-13-06032-f001:**
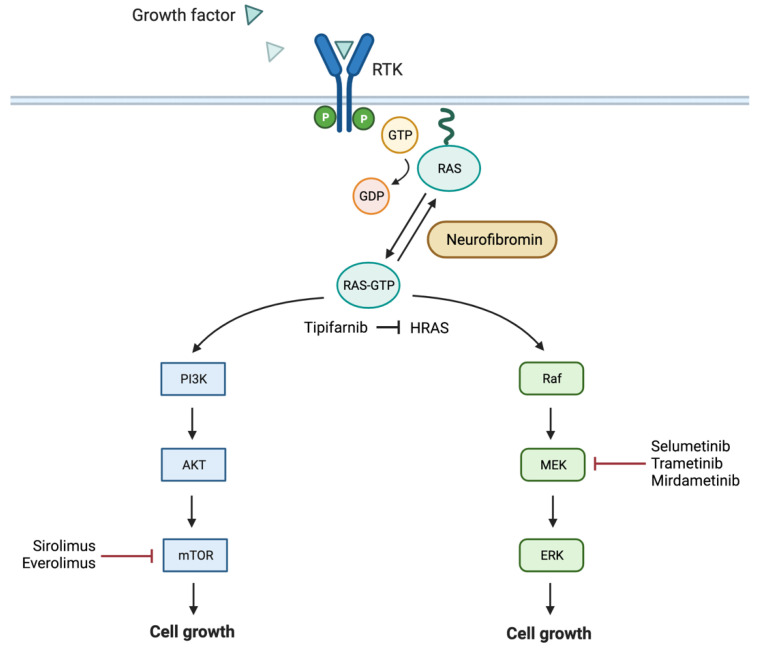
Schematic representation of the main NF1 associated signaling pathways with noted targeted therapies. RTK = receptor tyrosine kinase. Figure created with biorender.com (accessed on 19 October 2021).

**Figure 2 cancers-13-06032-f002:**
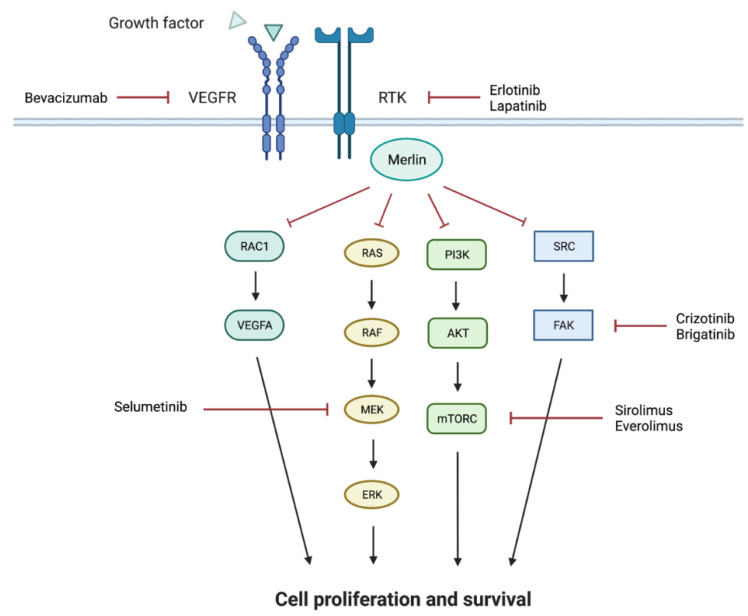
Simplified schema of the signaling pathways implicated in tumor formation with loss of merlin, the protein product of *NF2.* Therapies utilized to target these activated pathways are noted. This figure was created with Biorender.com (accessed on 19 October 2021).

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
