# Peer review of "Targeted Therapies for the Neurofibromatoses"

_cancers, 2021, doi:10.3390/cancers13236032_

Round 1

Reviewer 1 Report

The auhors present a rapid and well-described overwiew about the state-of-art of therapeutic treatment of neurofribromatoses-related tumors. There is a special focus on the new and ongoing oral treatment for plexiform neurofibromas or OPG. The review can contribute to expand this information also to clinicians not specifically involved in such diseases 

I have only some minor issues, mainly related to the surgical treatment of PNST (comprising malignant forms)

Par 2.2 (lines 266-267) "Therapeutic options for MPNST remain limited and currently center around surgical resection, with the goal being complete excision with negative margins". This is, currently, less true than in the past, because amputation not always has an impact on progression-free survival, but certainly it has a dramatic impact on residual quality of life. I suggest, aslo according to the most recent works about this topic, to modify this sentence introducing the concep of judicius and meticulous maximal safe resection (see i.e. Martin E. Morbidity and Function Loss After Resection of Malignant Peripheral Nerve Sheath Tumors. Neurosurgery. 2021 Sep 15:nyab342. doi: 10.1093/neuros/nyab342. AND Lu VM et al. The clinical course and role of surgery in pediatric malignant peripheral nerve sheath tumors: a database study. J Neurosurg Pediatr. 2021 Oct 8:1-8. doi: 10.3171/2021.7.PEDS21263.).

The maximal safe resection is a concept valid and feasible also for some huge plexiform neurofibromas, also thanks to some intraoperative adjunt as fluorescent dye, feasible not only in CNS tumors but also in PNST (see i.e. Vetrano IG et al. Fluorescein-guided removal of peripheral nerve sheath tumors: a preliminary analysis of 20 cases. J Neurosurg. 2019 Dec 6:1-10. doi: 10.3171/2019.9.JNS19970)

Finally, another question. Whereas RT plays a limited role both in PNST and for vestibular schwannomas or other CNS-NF related tumors, there is space for proton radiation therapy in treating such tumors?

Author Response

Reviewer 1:

We appreciate the reviewer’s enthusiasm for the review and thoughtful comments.  Please see our responses below. 

The authors present a rapid and well-described overview about the state-of-art of therapeutic treatment of neurofribromatoses-related tumors. There is a special focus on the new and ongoing oral treatment for plexiform neurofibromas or OPG. The review can contribute to expand this information also to clinicians not specifically involved in such diseases 

I have only some minor issues, mainly related to the surgical treatment of PNST (comprising malignant forms)

Par 2.2 (lines 266-267) "Therapeutic options for MPNST remain limited and currently center around surgical resection, with the goal being complete excision with negative margins". This is, currently, less true than in the past, because amputation not always has an impact on progression-free survival, but certainly it has a dramatic impact on residual quality of life. I suggest, aslo according to the most recent works about this topic, to modify this sentence introducing the concep of judicius and meticulous maximal safe resection (see i.e. Martin E. Morbidity and Function Loss After Resection of Malignant Peripheral Nerve Sheath Tumors. Neurosurgery. 2021 Sep 15:nyab342. doi: 10.1093/neuros/nyab342. AND Lu VM et al. The clinical course and role of surgery in pediatric malignant peripheral nerve sheath tumors: a database study. J Neurosurg Pediatr. 2021 Oct 8:1-8. doi: 10.3171/2021.7.PEDS21263.).

The maximal safe resection is a concept valid and feasible also for some huge plexiform neurofibromas, also thanks to some intraoperative adjunt as fluorescent dye, feasible not only in CNS tumors but also in PNST (see i.e. Vetrano IG et al. Fluorescein-guided removal of peripheral nerve sheath tumors: a preliminary analysis of 20 cases. J Neurosurg. 2019 Dec 6:1-10. doi: 10.3171/2019.9.JNS19970)

We appreciate the reviewer’s comments and agree. We have revised the sentence concerning surgical resection in the MPNST part of the manuscript and added the suggested references.  Use of fluorescent dye (and the suggested reference) has been added to the PN section of the manuscript.

Finally, another question. Whereas RT plays a limited role both in PNST and for vestibular schwannomas or other CNS-NF related tumors, there is space for proton radiation therapy in treating such tumors?

We appreciate this interesting question raised on the use of photon versus proton radiation for PNST and vestibular schwannomas.  At this time, there is limited published data on the use of proton radiation for either indication.  We have added an appropriate sentence to highlight the limited information.

Reviewer 2 Report

In this manuscript, Drs. Sanchez, Bui and Klesse review current therapeutic options for tumors associated with neurofibromatosis. These cancer predisposing syndromes are less studied due to their rarity, but nevertheless can cause significant morbidity and mortality. The review is well written and conveys a significant amount of information, and thus, will be a useful resource for many clinicians and researchers.

One minor suggestion is to discuss genomic, molecular, and preclinical studies that may lead to the next breakthroughs in the management of these syndromes and cancers. For instance, are there any mouse models or organoids for these syndromes? Are tumors derived from NF gnomically similar to their spontaneous counterparts, and are there any recurrent genetic and epigenetic changes in addition to the losses of NF1 and NF2? Are there any preclinical testing studies that may point to the next effective drugs? I think these discussions may give readers a better sense of the research directions in this field.

Another minor suggestion is to add molecular targets for the drugs listed in Table 1.

Author Response

In this manuscript, Drs. Sanchez, Bui and Klesse review current therapeutic options for tumors associated with neurofibromatosis. These cancer predisposing syndromes are less studied due to their rarity, but nevertheless can cause significant morbidity and mortality. The review is well written and conveys a significant amount of information, and thus, will be a useful resource for many clinicians and researchers.

We appreciate the reviewer’s comments.  Please see below for our responses. 

One minor suggestion is to discuss genomic, molecular, and preclinical studies that may lead to the next breakthroughs in the management of these syndromes and cancers. For instance, are there any mouse models or organoids for these syndromes?

We agree that this would be beneficial information for the readers.  To this end, we have expanded the discussion of the preclinical testing consortium in the conclusions to specifically include the use of animal models for target selection. 

Are tumors derived from NF gnomically similar to their spontaneous counterparts, and are there any recurrent genetic and epigenetic changes in addition to the losses of NF1 and NF2?

We appreciate this interesting suggestion from the reviewer.  Although histologically similar (and often with similar pathways implicated in tumor development) the goal of this review in particularly was to highlight the unique biological behavior of NF associated tumors and the need for specific NF targeted therapy.  Where appropriate recurrent changes are known (such as loss of CDKN2A) in NF are outlined in those sections. 

Are there any preclinical testing studies that may point to the next effective drugs? I think these discussions may give readers a better sense of the research directions in this field.

We agree with this suggestion by the reviewer and have expanded the discussion of the Synodos and preclinical testing consortium to highlight this ongoing work. 

Another minor suggestion is to add molecular targets for the drugs listed in Table 1.

As per the reviewer’s suggestion, we have added the molecular target to Table 1.